# Preneoplastic Low-Risk Mammary Ductal Lesions (Atypical Ductal Hyperplasia and Ductal Carcinoma In Situ Spectrum): Current Status and Future Directions

**DOI:** 10.3390/cancers14030507

**Published:** 2022-01-20

**Authors:** Thaer Khoury

**Affiliations:** Department of Pathology, Roswell Park Cancer Institute, Elm & Carlton Streets, Buffalo, NY 14263, USA; thaer.khoury@roswellpark.org

**Keywords:** preneoplastic breast disease, noticed ductal carcinoma in situ, atypical ductal hyperplasia, active surveillance trials, de-escalation

## Abstract

**Simple Summary:**

Overtreatment which means providing a patient with unnecessary medical intervention has recently become the center of attention in breast diseases. One of these lesions is low-risk ductal carcinoma in situ (DCIS), for which the standard of treatment is wide local excision/mastectomy, with or without radiation therapy, with or without hormonal therapy. There are many worldwide ongoing clinical trials aiming to de-escalate this therapy by examining whether active surveillance (AS) is sufficient. These trials vary in pathology inclusion and exclusion criteria. There is lack of consensus on the definition of low-risk DCIS, in particular the overlap with atypical ductal hyperplasia (ADH), DCIS grading, and lack of definition of comedonecrosis. This review discusses AS trial from pathology standpoint and provides few recommendations.

**Abstract:**

Intraepithelial mammary ductal neoplasia is a spectrum of disease that varies from atypical ductal hyperplasia (ADH), low-grade (LG), intermediate-grade (IG), to high-grade (HG) ductal carcinoma in situ (DCIS). While ADH has the lowest prognostic significance, HG-DCIS carries the highest risk. Due to widely used screening mammography, the number of intraepithelial mammary ductal neoplastic lesions has increased. The consequence of this practice is the increase in the number of patients who are overdiagnosed and, therefore, overtreated. The active surveillance (AS) trials are initiated to separate lesions that require active treatment from those that can be safely monitored and only be treated when they develop a change in the clinical/radiologic characteristics. At the same time, the natural history of these lesions can be evaluated. This review aims to evaluate ADH/DCIS as a spectrum of intraductal neoplastic disease (risk and histomorphology); examine the controversies of distinguishing ADH vs. DCIS and the grading of DCIS; review the upgrading for both ADH and DCIS with emphasis on the variation of methods of detection and the definitions of upgrading; and evaluate the impact of all these variables on the AS trials.

## 1. Introduction

Atypical ductal hyperplasia (ADH) belongs to the low-grade pathway of estrogen receptor (ER)-positive/luminal type invasive breast carcinoma (iBC) [1], and it is one of the earliest steps in this pathway [2]. Ductal carcinoma in situ (DCIS) is a noninvasive form of ductal carcinoma. Both ADH and DCIS are characterized by intraepithelial ductal neoplastic proliferation confined to the basement membrane. The incidence of ADH/DCIS dramatically increased after the introduction of population-based screening mammography but remains steady in more recent years [3,4]. The distinction between these two entities is somewhat murky with arbitrary cutoffs that had nothing to do with biology, instead having the least interobserver variability [5,6].

With the increase in the detection of these lesions, the number of surgical excisions has increased in parallel. The prognosis of these lesions together with the early-stage ER-positive/HER2-negative low-grade iBC is excellent [7]. Therefore, a fundamental review of clinical practice and approach for these lesions has taken place; specifically, a deescalating of the current standard of treatment for DCIS and low-risk iBC. Currently, the standard of treatment includes surgical resection with possible radiation therapy (RT) and/or hormonal therapy (HT) [8,9,10,11,12,13]. However, these treatments are viewed as extreme for at least a subset of patients with unnecessary subsequent toxicity and increased cost [14]. Therefore, there are important opportunities to tailor therapy for these patients and at the same time study the natural history of these lesions.

There are at least four ongoing international clinical trials aimed at studying the natural history of low-risk DCIS and defining specific clinical/pathological biomarkers for a subset of patients who may benefit from adjuvant therapies [15,16,17,18]. They all have a similar approach with performing active surveillance (AS) and differ in few details.

In this article, the author presents: (1) Review of current AS trials, including enrollment criteria, overlaps, and controversies; (2) Overview of the spectrum of ductal lesion in terms of definition, histomorphology, biology, clinical presentation and significance; (3) Review of retrospective studies that simulate AS trials; (4) Discussion of how the pathologist plays a major role in these trials, particularly when it comes to differentiating ADH from low grade (LG)-DCIS and LG-DCIS from high-grade (HG)-DCIS; and (5) Integration of this information together and recommendation of a certain approach for the AS trials upon completion.

## 2. Active Surveillance Trials

There are at least four AS trials worldwide, including Comparison of Operative to Monitoring and Endocrine Therapy (COMET) in the United States [15], the low risk DCIS trial (LORIS) in United Kingdom [16], the “LOw Risk Dcis” (LORD) in the Netherlands [17], and LORETTA in Japan [18]. They differ in the inclusion criteria and design, but they share goals which are: (1) Identifying a subset of patients who could be spared from overtreatment and (2) Evaluating the natural history of low-risk DCIS (Table 1).

There have been few amendments since the start of the trials. The COMET trial amended the inclusion criteria to add patients whose surgical resection margin was positive for DCIS, and patients with a diagnosis of ADH bordering on DCIS. They also removed comedonecrosis as one of the exclusion criteria. LORIS amended the inclusion criteria to include cases with a diagnosis that is given by a local pathologist as DCIS and downgraded to ADH by the central review. LORD originally, in 2017, included only LG-DCIS. However, the rate of accrual was very low. Therefore, in 2020, they amended the inclusion criteria to include patients with LG- or intermediate grade (IG)-DCIS. Patient preference was also considered, but to decrease the number of patients who might have high grade (HG)-DCIS, they required that the DCIS be ER-positive/HER2-negative.

The differences between these trials make the comparisons challenging. Mass lesion is included in LORETTA but excluded by the other three trials. Mass lesion is known to have high-risk of upgrading for either DCIS or ADH [19,20,21,22]. Comedonecrosis is another variable that has been debated as being a risk factor [23]. Moreover, the definition of comedonecrosis is not agreed on [24]. This variable is included in the amended COMET but excluded by LORTETTA. LG- or IG-DCIS are included in all trials but LORD, which included cases with LG-DCIS only. However, three years into the trial, it was amended to include patients with IG-DCIS. Nuclear grade might also play a role in the underestimation of iBC and/or HG-DCIS [23]. Hormonal therapy is optional in COMET, mandatory in LORETTA, and not possible in LORIS or LORD. Hormonal therapy is known to decrease the risk of developing ipsilateral breast cancer in both DCIS and ADH [9,12,13,25,26,27]. A fifth study, LARRIKIN, has been conducted in Australia and New Zealand, with more stringent criteria. The inclusion criteria are women older than 55 years; DCIS detected by screening mammography or incidentally; the size of calcifications less than 25 mm; and, histologically, DCIS has to be ER-positive /HER2-negative with no comedonecrosis [28].

## 3. Atypical Ductal Hyperplasia and Ductal Carcinoma In Situ Spectrum-Overview

Atypical ductal hyperplasia and LG-DCIS have similar histomorphology, but they differ in the extent of disease. In this section, an overview of the controversy of separating these two diagnoses is discussed.

### 3.1. Atypical Ductal Hyperplasia-Overview

Atypical ductal hyperplasia is a preneoplastic mammary ductal lesion. It is mostly diagnosed in a mammogram screening setting in which microcalcifications are identified. The incidence of ADH remained stable after the implementation of population-based screening mammography, accounting for 2–14% of diagnoses [4,29,30,31]. The relative risk of developing iBC is between three and five times [32]. Amin et al. followed up 71 patients who had ADH detected in screening mammography, treated with wide local excision, and had no upgrading (to DCIS or iBC) for a median time of 71.3 months. Nine (12%) patients developed subsequent cancer (four DCIS and five iBC), seven ipsilateral and two contralateral [33].

Histologically, ADH is composed of intraductal neoplastic epithelial proliferation that involves terminal ductal lobular units. The cells are small, round, and monomorphic with well-defined borders. The cells are arranged in complex atypical growth pattern including solid, cribriform or micropapillary. In the cribriform pattern, cells are polarized around the lumens. In the micropapillary pattern, the tips of the micropapillae are broader than the bases. Part of the involved ducts could show usual ductal hyperplasia (Figure 1).

### 3.2. Ductal Carcinoma In Situ—Overview

Ductal carcinoma in situ is a nonobligate precursor of iBC, defined as noninvasive proliferation of neoplastic ductal epithelial cells confined to the basement membrane, involving terminal ductal lobular units. Approximately 85% of DCIS cases are estimated to be detected mammographically, typically as microcalcifications [34]. Like ADH, the rate of DCIS has increased due to the implementation of population-based screening mammography [3,4], accounting for about 15% of all breast carcinomas in western countries [35].

There is evidence that LG- and HG-DCIS follow different pathways and have different molecular and genetic profiles. LG-DCIS, like ADH, belongs to the low-grade pathway of ER-positive/luminal type iBC [1]. HG-DCIS has more genetic complexity and more frequent genomic alterations than LG-DCIS [36].

### 3.3. Severe Atypical Ductal Hyperplasia Bordering on Ductal Carcinoma In Situ

Both ADH and LG-DCIS have similar histomorphology, but they differ in the extent of the disease, except when the spaces are partially involved with atypical cells, then the diagnosis is ADH regardless of the extent of the disease. There have been multiple attempts to separate ADH from LG-DCIS. These attempts were made only for practical reasons and had nothing to do with the biology of the diseases or their clinical outcomes. Page et al. originally defined LG-DCIS as tumor cells involving at least two spaces [5]. Subsequently, Tavassoli and Norris proposed a cutoff of more than 2 mm to render a diagnosis of LG-DCIS [6]. For some time, a combination between these two criteria (two spaces and size of more than 2 mm) has been adopted to separate ADH from DCIS. There are a few challenges in applying these criteria; first, measuring the extent of the disease is challenging in certain situations, when the involved spaces are partially complex and partially flat (e.g., flat epithelial atypia) (Figure 1A) or partially atypical and partially hyperplastic (Figure 1B); second, there is high interobserver variability in defining the nuclear grade; and third, the presence of mixed grades within the same case or within the same duct.

Due to the risk of overtreatment, the world health organization (WHO) working group recommended a more conservative approach, particularly in small and limited material such as core needle biopsy (CNB) [37]. Moreover, some authors accept lesions as large as 3–4 mm as ADH (Figure 2A,B) [38]. In the author’s opinion and the opinion of many breast pathologists, these pathologic features are acceptable to designate a lesion as severe ADH bordering on DCIS including a single lesion that measures 3–4 mm, mixed low and intermediate grade nuclei, or small size of comedonecrosis (Figure 3). The extent of the disease could be evaluated on the subsequent surgical excision. About 30% of such cases (unpublished data from the author) had no residual disease in subsequent excision. Therefore, these patients would have been designated as having DCIS and possibly received unnecessary treatment such as RT if the lesion was designated as DCIS on the CNB.

## 4. Upgrading

There have been many studies evaluating the rate of upgrading. Upgraded lesions are defined as lesions initially diagnosed as ADH in CNB but are found to harbor DCIS or iBC at excisional biopsy, or DCIS in CNB and are found to harbor iBC at the excisional biopsy. These studies suffer from many variabilities; they are all retrospectives; have variable source settings (cancer center, general hospital, etc.), variable clinical settings (screening vs. non-screening), variable imaging modalities (mammography, ultrasound (US), magnetic imaging resonance (MRI)), and more importantly the variation in the definition of upgrading. While upgrading to any grade DCIS and/or iBC is clinically relevant to treatment purposes, upgrading to HG-DCIS and/or iBC is relevant to the AS trials.

One of the arguments for the AS trials is the low incidence of upgrading to iBC and/or HG-DCIS. In this section, studies that evaluated the upgrading rate for both ADH and low-risk DCIS are discussed.

### 4.1. Upgrading of Atypical Ductal Hyperplasia and Ductal Carcinoma In Situ—Overall Studies

#### 4.1.1. Upgrading from ADH to Any Grade-DCIS and/or Invasive Carcinoma

Schiaffino et al. performed a meta-analysis on 6458 cases from 93 studies and explored the risk of upgrading for ADH with emphasis on radiology confounders [39]. A wide range of upgrading rates were reported ranging from 0% [40] to 84% [41]. The main reason for this wide range of upgrading rate was the heterogeneity among the studies, including the type of guidance and biopsy clipper. Although 93 studies filtered from 530 studies were included in this meta-analysis, the overall heterogeneity was high (*I*^2^ = 80%; >50% was considered high) [39]. There is a potential for higher heterogeneity if pathological confounders were included, as the inter-/intra-observer variabilities in defining ADH/DCIS are high [42,43].

The acceptable imaging modality for the COMET, LORIS, and LORD trials is screening mammography, while for LORETTA, it is any imaging modality (Table 1). Therefore, abstracting the rates of upgrades using mammography as the imaging modality would be relevant to the first three AS trials. The overall rate of upgrading using stereotactic guidance was 23% (18% to DCIS and 5% to iBC) with heterogeneity of *I*^2^ = (63%, 60% and 42%, respectively). The rate of upgrading with US or MRI was observed to be higher than stereotactic guidance, 48% and 32%, respectively, and upgrading to iBC was 22% and 10%, respectively [39]. The reason for the higher upgrading rate for US-detected lesions is that these lesions are usually mass-forming [44]. MRI-detected lesions are usually performed for high-risk patients or for staging purposes on patients who already have a diagnosis of carcinoma in the same or in the contralateral breast [45]. Therefore, LORETTA trial might have a higher number of underestimations of this disease than the other trials when the imaging modality is considered.

#### 4.1.2. Upgrading from Ductal Carcinoma In Situ to Invasive Carcinoma

The upgrade of DCIS to iBC is another concern regarding this disease, with an overall rate estimated at approximately 20% [46].

Brennan et al. performed a meta-analysis to report the estimates of the upgrade rate for DCIS detected by CNB to iBC in the subsequent surgical excision. They identified 7350 DCIS cases from 52 studies. There were 1736 cases with iBC on the subsequent excision with random-effects pooled estimate of 25.9% (95% confidence interval: 22.5%, 29.5%). They identified several preoperative variables that were significantly associated with a higher upgrading rate in the univariate analysis, including the automated 14-gauge device, HG-DCIS, a larger lesion size (larger than 20 mm) on imaging, a (BI-RADS) score of 4 or 5, mammographic mass, and palpability. However, multivariate analysis was not performed. Interestingly, the rate of upgrading was found to be lower in older studies [22].

Nicosia et al. found that the upgrade rate was significantly lower (8.2%) in patients who showed complete mammographic removal of the lesion. They concluded that the rate of upgrading is low enough to recommend AS instead of surgical removal. Moreover, they included all grades of DCIS, which means this ratio may be less if HG-DCIS cases were excluded [47].

### 4.2. Upgrading of Atypical Ductal Hyperplasia and Ductal Carcinoma In Situ—Studies Simulating Active Surveillance Trials

One of the arguments against AS trials is that it is unethical to leave a clinically significant disease such as iBC or HG-DCIS without definitive treatment. However, not including potential high-risk cases would undermine the power of the studies, making them unfeasible. Those who develop iBC (primary endpoint) at a certain point of follow-up may have had iBC at the time of enrollment not due to the nature of DCIS. Some of these studies examined the association of pathological and radiological variables with upgrading.

#### 4.2.1. Atypical Ductal Hyperplasia

A single study by the author was conducted that simulated the COMET trial. The included cases (*n* = 165) were a spectrum of ADH, ADH bordering on DCIS, and low to intermediate grade DCIS with no comedonecrosis. The cases were reclassified upon reviewing dichotomously into ADH versus DCIS. There was a total of 9 (5.5%) upgraded cases, one to HG-DCIS, five to iBC, and four to DCIS with comedonecrosis. The iBC cases were all low-risk; the corresponding CNB had ADH diagnosis in three cases and DCIS in two. The authors recommended adding ADH and ADH bordering on DCIS to existing AS trials [48]. Farshid et al. published a study somewhat similar, but with less stringent criteria. They included 114 cases with a diagnosis of ADH. The upgrading rate to high-risk lesion (HG-DCIS and/or iBC) was 13.6%. There are a few reasons for this relatively high upgrading rate. They included cases with any mammographic abnormality (microcalcifications ranging from 3 to 150 mm, or mass) and all BIRADS scores. The iBC cases were all less than 10 mm, node negative, and ER-positive. Two cases had *HER2* amplification by in situ *hybridization* [49].

#### 4.2.2. Ductal Carcinoma In Situ

Few retrospective studies have simulated AS trials (Table 2). The upgrading rate to iBC was comparable between the COMET and LORIS trials, ranging from 6% to 22% and from 0% to 24%, respectively. The upgrading rate in the LORD trial ranged from 0% to 10%. The average upgrading rate for COMET, LORIS and LORD trials was 22 of 474 (8%), 113 of 979 (11.5%) and 6 of 111 (5.4%), respectively (Table 2) [23,48,50,51,52,53,54,55,56]. It is worth noting that these results were published before amending the LORD trial. The original LORD trial included only LG-DCIS; then, they amended the inclusion criteria to include IG-DCIS or patient preference. A single study classified the cases based on the amended LORD trial and found the risk of upgrading 25% [57]. Chavez de Paz Villanueva et al. reviewed records from the National Cancer Database for 37,544 women with a diagnosis of non-HG-DCIS, 22.2% of whom had upgrading to iBC. The upgraded iBC were more likely to be smaller and lower grade. Multivariate logistic regression analysis demonstrated that younger age, ER-negative, treatment at an academic facility, and higher annual income were significantly associated with upgrading to iBC. The final pathological stage for the iBC cases was as follows: 9.9% stage 2, 1.24% stage 3 and 0.2% stage 4. Adverse biology was identified in 12% (ER-negative) and at least 6.5% (HER2-positive; 24.5% of the cases did not have HER2 status reported) [58]. However, these results should be interpreted with caution for many reasons, most importantly because the AS trials inclusion and exclusion criteria were not accounted for in this study. Moreover, there is evidence that the interobserver variability is high in distinguishing IG-DCIS from HG-DCIS [59,60,61,62].

Only a few studies evaluated the factors associated with the risk of upgrading. Intermediate grade was found to associate with upgrading to iBC in two studies [50,52]. Two studies by the author included women who met the amended COMET trial criteria. The first study evaluated the risk of upgrading to iBC and/or HG-DCIS and found that women with larger mammographic calcification spans had increased rate of upgrading with the best cutoff 1.1 cm. When the risk of upgrading was defined as iBC, the increased number of spaces with comedonecrosis and increased percentage of ductal diameter occupied by necrosis were statistically significant. The best cutoff of percentage of necrosis that correlated with upgrading was 53.5% [23]. In another study on the same group of women, touching tumor-infiltrating lymphocytes (TILs) were evaluated. When upgrading was defined as HG-DCIS, the increase in the average, or the highest number of touching TILs, and IG-DCIS associated with upgrading. A logistic regression model was created combining TILs and IG-DCIS and had an accuracy of 0.775 [56].

Pilewskie et al. reported the highest rate of upgrading to iBC of 20%, which is almost twice as much as any other study. It is noted that the CNB cases were not reviewed by a pathologist, and the nuclear grade was rather abstracted from the pathology report [52]. Given the high rate of interobserver variability in grading DCIS, and the fact that the nuclear grade correlates with the upgrading to iBC, it is difficult to interpret these results. In addition, patients who were upgraded to iBC were found to be more likely to have IG-DCIS on CNB and to have had a mastectomy. Mastectomy is normally reserved for DCIS patients who have extensive disease. Podoll et al. found that among 105 cases with upgrading to iBC, three (3%) met the eligibility criteria for the LORD trial and 20 (19%) for the LORIS trial. However, to calculate the ratios, the total number of cases with or without upgrading that met the eligibility criteria for either trial should have been mentioned. Therefore, calculating the rate of upgrading is not possible [63].

One of the aims of the AS trials is to investigate the natural history of low-risk lesions. Therefore, leaving a high-risk lesion without definitive treatment would interfere with the final interpretation of the data. High-risk lesions are defined as tumors with positive lymph node or have adverse biology such as HER2-positive (≥8-mm) [64] or triple negative (>5-mm) [65]. However, the way the results were listed in these retrospective studies makes it difficult to identify the high-risk lesions. In total, 3 of 58 (5.2%) patients who would be eligible for LORIS trial had at least one positive lymph node [52]. All patients who would be eligible for COMET, LORIS or LORD trials had not a single positive lymph node [55]. In a study by the author, there were 12 iBC, none of which was high-risk [23]. In another study by the author, all five patients who had upgrading had low-risk disease [48]. Podoll et al. found that at least 2 of 20 patients who met the eligibility criteria for LORIS and/or LORD had high-risk lesion in the subsequent excision, both with positive lymph nodes [63]. Iwamoto et al. found that among the patients who were eligible for LORIS trial, two had triple negative iBC and two HER2-positive in the subsequent excision. However, the sizes of the corresponding tumors were not mentioned [57]. The conclusion is that in the ongoing AS trials, there must be a small but considerable number of patients who already have high-risk disease at the time of enrollment.

### 4.3. Nomograms and Logistic Regression Analyses

For practical reasons and to aid patients and treating physicians in the decision making regarding the treatment of ADH/DCIS detected on CNB, there have been some studies that constructed risk factors in logistic regression models. These models could serve as a reference to calculate the risk of upgrade.

#### 4.3.1. Atypical Ductal Hyperplasia

There are few studies that constructed the risk factors associated with upgrading in a nomogram format. The first nomogram published by the author included patients with mammography-detected lesions. Upgrading was defined as any type of DCIS or iBC in the subsequent surgical resection. The following variables were included in the nomogram: age, menopausal status, HT, history of breast cancer, number of involved cores, solid histologic growth pattern, size of the lesion, and type of lesion (mass versus microcalcifications). The value of the area under the curve (AUC) was 0.775. The risk of upgrading could be as low as 8% if the total points were ≤49.41. A limiting factor to this study, however, was the lack of validation cohort [20]. Huang et al. created a nomogram with a similar approach but for US-detected lesions. The cases studied were divided into training set and validation set. The following variables were constructed in the nomogram including age, mass palpation, calcifications on US, ADH extent, and suspected malignancy. The AUC values were 0.783 in the training set and 0.753 in the validation set [44]. The author concludes that, in addition to the clinical use, this risk modeling approach could be useful in assessing the results of the AS trials.

#### 4.3.2. Ductal Carcinoma In Situ

There have been several studies that performed logistic regression analyses on several significant variables to identify certain variables that can be reliable to certain extent dividing the cases into low-risk for upgrading vs. high-risk. However, these studies included HG-DCIS. Since HG-DCIS is not included in the AS trials, this discussion is beyond the scope of this review.

## 5. Ductal Carcinoma In Situ

### 5.1. Histomorphology and Grading

Morphologically, the involved ductal lobular units have neoplastic ductal epithelial proliferation arranged in a complex architecture. The architecture could be cribriform (with fenestrations) (Figure 4A), solid (Figure 4B), micropapillary (club-shaped) (Figure 4C), papillary (Figure 4D), comedo (Figure 4E), or clinging (single layer), (Figure 4F).

There are many grading systems for DCIS [66]; the most commonly used are the consensus conference on the classification of DCIS [67]; Lagios’ nuclear grading systems for DCIS [68]; DCIS grading according to the College of American Pathologists [69]; and the United Kingdom’s National Health Service/breast screening program guidelines for grading DCIS.

The WHO classifies DCIS into three grades (low, intermediate, or high). Low-grade DCIS is composed of small monomorphic cells arranged in a complex growth pattern (cribriform, micropapillary, or less often solid). The nuclei are uniform in size and shape with irregular chromatin and indistinct nucleoli. The nuclei measure 1 to 1.5 times the size of an erythrocyte. Mitotic figures are rare. The lesion involves more than two complete spaces (or measuring >2-mm) (Figure 5A). However, as mentioned above, small lesions (up to 3 or 4 mm) could be classified as severe ADH bordering on DCIS. Intermediate-grade DCIS is composed of cells that show moderate variability in size, shape, and polarization. The nuclei have variably coarse chromatin with sometimes prominent nucleoli. Mitoses may be present. Necrosis (either focal or comedo) may be seen (Figure 5B). High-grade DCIS is composed of large, atypical cells. Unlike LG- or IG-DCIS, polarization around the lumen is minimal. The nuclei are large, pleomorphic, with irregular contours, coarse chromatin, and often prominent nucleoli. The nuclei size of the nuclei are >2.5 times the size of an erythrocyte. Mitoses are usually frequent. Central comedonecrosis is usually present. The most common growth pattern is solid. Cribriform, micropapillary, or flat (also known as clinging) can also be seen (Figure 5C) [34].

### 5.2. Natural History and Prognostic Tools for Ductal Carcinoma In Situ

The natural history of DCIS is unclear, owing to the clinical practice in which a diagnosis of any form of DCIS is routinely excised. There are only a few studies that evaluated the natural history of DCIS by reviewing those who declined surgical removal or were misdiagnosed as benign. In a small study population, up to 50% of DCIS has been reported to progress to iBC over a 30-year period [70,71]. Maxwell et al. reported that 33% of women who had a diagnosis of DCIS with no subsequent surgical removal developed iBC after a median and range of follow-up 45 (12–144) months, 48% from HG-DCIS, 32% from IG-DCIS, and 18% from LG-DCIS cases [72]. In a Surveillance Epidemiology and End Results (SEER) study applied a similar approach. A high variability was found in the progression rate to iBC with a 10-year risk ranging from 15% to 28% [73]. Complex approaches to answer the question of the natural history of DCIS have been investigated. One of these approaches was to employ population-based models of incidence and progression in conjunction with breast cancer incidence data from the SEER program. Chootipongchaivat et al. found that between 65.8% and 100% of DCIS progressed to iBC. However, the estimated proportion of DCIS with overdiagnosis ranged from 3.1 to 65.8%. In addition, these models did not provide pathological characteristics [74].

Defining the prognosis of DCIS after surgical removal has been an area of extensive research. Few models have been developed, including Memorial Sloan-Kettering Cancer Center (MSKCC) [75], Oncotype [76,77], and University of South Carolina Van Nuys Prognostic Index (USC-VNPI) [78]. While Oncotype examines DCIS at the molecular level, MSKCC and USC-VNPI use clinical and pathological variables to predict outcomes. Although the DCIS Oncotype score correlates well with the 10-year risk of local recurrence of LG-DCIS, IG-DCIS, and HG-DCIS [76,77], other clinical and pathologic variables are more predictive [79].

## 6. Treatment

### 6.1. Atypical Ductal Hyperplasia

The National Comprehensive Cancer Network (NCCN) guidelines recommend surgical removal of the ADH lesion [80]. However, if the risk versus benefit of surgery comes into question, the treating physician can use published literature for guidance. Two nomograms have been published (see above), one for cases detected by screening mammography and one for cases detected by US [20,44]. Caplain et al. developed institutional guidelines that ADH does not require to be excised if it is (a) <6 mm in size and completely removed or (b) <6 mm in size and incompletely removed but with <2 foci. Then, they prospectively reviewed 41 cases which were excised contrary to the above guidelines and found only one (2%) was upgraded, while those that were excised following these guidelines had an upgrade rate of 37% [81].

Since most of the iBC cases that develop in women with ADH are strongly positive for ER, it is logical to use selective estrogen receptor modulators (SERM) and aromatase inhibitors (AI) as a preventive measure. Several large breast chemoprevention clinical trials have shown risk reduction ranging from 41% to 79% [27]. A clinical trial by the national surgical adjuvant breast and bowel project (NSABP) found that tamoxifen reduced the risk of developing breast carcinoma in 86% of women who had ADH versus placebo [13]. However, this chemoprevention approach is rarely prescribed or used [27].

### 6.2. Ductal Carcinoma In Situ

The standard of treatment of DCIS is surgical removal of the lesion with or without RT, with or without HT. The extent of surgery (wide excision versus mastectomy) depends on the extent of the disease [8]. Mastectomy is indicated if the area of DCIS cannot be removed by oncologically acceptable excision (preserving cosmetically acceptable breast), multicentricity, or if the patient cannot undergo RT. If mastectomy is chosen, a sentinel lymph node biopsy is required, as about 20% of DCIS is upgraded to iBC [46]. Mastectomy precludes subsequent sentinel node biopsy. Radiation therapy reduces the risk of recurrence by 50%. However, it does not significantly change the overall survival of DCIS patients [9,10,11]. Adjuvant HT with tamoxifen associates with reduced risk of recurrence [9,12,13,25,26]. Although these therapies reduce the recurrence rates, they do not come without side effects, some of which are serious, such as angiosarcoma post-RT [82], and deep venous thrombosis post-HT [83].

## 7. Challenges in the Active Surveillance Trials from Pathology Standpoint

The inclusion criteria for COMET trial includes LG- and IG-DCIS with any degree of necrosis. LORIS trial includes LG-DCIS or lower half of IG-DCIS with no comedonecrosis. LORD trial accepts LG-DCIS or IG-DCIS with ER-positive/HER2-negative. LORETTA includes LG-DCIS or IG-DCIS with no comedonecrosis and must be ER-positive/HER2-negative. In most of these studies, a pathologist or two must evaluate these variables and decide the eligibility of the patient for any given study. Two major areas for inconsistency, and, therefore, non-uniformity in the recruited cases among the trails, are DCIS grading and defining comedonecrosis.

### 7.1. Challenges in Grading Ductal Carcinoma In Situ and Their Consequences on Active Surveillance Trial

As mentioned above, there are many grading systems for DCIS [66]. Moreover, cells with various nuclear grade could coexist in the same DCIS case (Figure 6A) with HG-DCIS (Figure 5B) and IG-DCIS (Figure 6C) and possibly within a single duct (Figure 7) [84]. This variation reflects the genetic heterogeneity of tumor cells [85]. The many grading systems and the variation of the nuclear grade in a single case are the main two reasons for the high interobserver variability [59], even when a two-tier nuclear grading was used [60]. Sneige et al. investigated the interobserver variability among six pathologists who evaluated the nuclear grade according to Lagio’s criteria. Only 35% of the cases had complete agreement. The kappa distinction between LG- and IG- and between IG- and HG- were fair and moderate, respectively [61].

#### 7.1.1. Downgrading Low-Risk Ductal Carcinoma In Situ to Atypical Ductal Hyperplasia or Atypical Ductal Hyperplasia Bordering on Ductal Carcinoma In Situ

As mentioned above, the diagnosis of some cases falls in between ADH and DCIS. The WHO recommends a more conservative approach. Following WHO recommendations, these cases are designated as severe ADH bordering on DCIS and treated conservatively with local excision. Paradoxically, if the case is designated as low-risk DCIS, the patient would be eligible for the AS trial and could avoid surgery. The COMET trial circumvented this issue by amending the study to include cases with severe ADH bordering on DCIS, but not ADH [15]. The LORIS trial accepted cases that were designated as DCIS which were downgraded by central review as ADH [16].

Tozbikian et al. found that most cases with a diagnosis of ADH bordering on DCIS were diagnosed as ADH after a review by a group of pathologists. With a median follow-up of 57 months, 3.8% developed subsequent ipsilateral carcinoma; interestingly, all in the ADH group [43]. Due to the lack of concrete diagnostic criteria for ADH, the interobserver variability could reach up to 60% [86,87,88], and the intraobserver variability is about 50% for atypia [89]. This would make an argument to call for including cases with ADH in the AS trials [48].

#### 7.1.2. Low-Risk Ductal Carcinoma In Situ Versus High-Grade Ductal Carcinoma In Situ

Distinguishing low-risk from HG-DCIS is crucial in the AS trials setting. If a HG-DCIS is mistakenly designated as low-risk and the patient is enrolled in the AS trial, it would have a negative impact for two reasons; first, the aim of studying the natural history of low-risk DCIS would be compromised; second, the patient would be left at risk for not being treated for a significant disease such as HG-DCIS or possibly concurrent iBC [22].

A recent study that included more than 5000 cases found that the proportion of LG-DCIS varies from 6% to 24% by department [62]. This means that the LORIS trial (before the amendment) may recruit patients from one center four times more than another, and the question always remains as to whether these cases are truly low grade.

Machine learning could assist in overcoming the variation in grading DCIS. Two studies from the same group were performed on the same group of DCIS cases. In the first, DCIS was graded manually and there was no correlation with the clinical outcomes [90]. In the second, DCIS was graded using quantitative image cytometry and achieved good correlation with clinical outcomes [84]. Although machine learning appears to be the solution for this issue, more work is still needed. The computer performs the scoring based on the training by humans following a set of criteria adopted by a given group. Therefore, while intraobserver variability is no longer a problem, inter-observer (inter-system) variability remains an issue. Therefore, the author recommends having an international consensus that agrees on a set of criteria and training a computer system to perform the grading.

### 7.2. Comedonecrosis Controversy and Its Consequences on Active Surveillance Clinical Trials

There is controversy on the definition of comedonecrosis. Comedo type DCIS was included in the 1997 classification as a subtype of DCIS besides cribriform, solid, micropapillary and papillary [67]. Then, the term “comedo” was used to describe central and expansile necrosis in any of the other four types [91]. CAP requires reporting the type of necrosis, comedo, punctate or none [69]. In a study by Harrison et al. breast pathologists received diagrams showing varying degrees of necrosis from 10% to 80% of the ductal diameter. The ratio that had highest agreement was 30%, though only one third of the pathologists surveyed chose it [24]. To circumvent this issue, the COMET trial removed comedonecrosis altogether as one of the exclusion criteria [15]. Since comedonecrosis is one of the exclusion criteria for the LORETTA and LORIS studies, it is expected that these studies would suffer from a non-uniformity of the enrolled cases.

The author examined whether comedonecrosis correlated with the upgrading to iBC. In this study, 129 cases that met the COMET trial criteria were investigated. The degree of comedonecrosis was measured by calculating the percentage of the ductal diameter occupied by necrosis, using image analysis. The degree of comedonecrosis correlated with the upgrading rate to iBC, with the best cutoff 53.5% [23].

The role of comedonecrosis in the natural history of DCIS will be better evaluated by the COMET trial, but not by LORIS or LORETTA. However, the caveat is that a good proportion of these patients would already have iBC at the time of diagnosis.

## 8. Conclusions

There is significant overlap between ADH, LG-, IG- and HG-DCIS in terms of definition, histologic features/criteria, biology, and clinical significance. Distinguishing these lesions from each other has become more important with the introduction of the AS trials, particularly between ADH and LG-DCIS and between LG-DCIS and HG-DCIS. The AS trials aimed to identify a subgroup of patients that could be monitored safely. Therefore, they targeted a component of this spectrum and designated it as low-risk DCIS which was defined as ER-positive LG- or IG-DCIS (with or without comedonecrosis, depending on the trial). However, they omitted ADH, which has the lowest risk of developing breast cancer. Although the studies differ in inclusion criteria, it is a good opportunity to evaluate different variables. As for the trial enrollment, the difficulty in recruiting is understandable, as the decision for a woman to excise or observe is not simple to make. The trials will be able to objectively evaluate some histological, radiological, and clinical variables that could narrow down the possibility of having residual high-risk disease such as iBC or HG-DCIS. Possible variables suggested by the author are nuclear grade, comedonecrosis, extent of calcifications, mass vs. calcifications, and TILs. Additional markers such as HER2 and possibly molecular studies could also be useful.

## Figures and Tables

**Figure 1 cancers-14-00507-f001:**
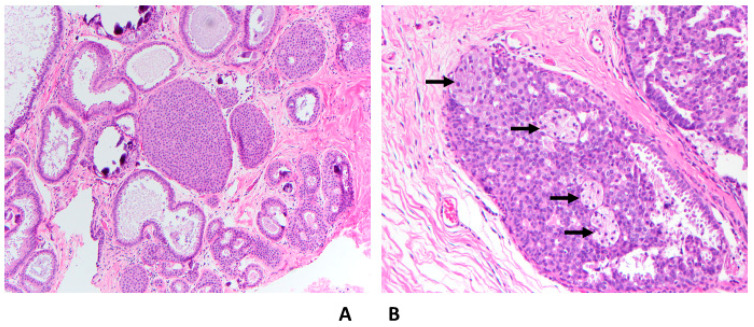
Atypical ductal hyperplasia; (**A**) Mixed ADH (solid and cribriform types) with flat epithelial atypia and associated macrocalcifications (H&E 10×); (**B**) Mixed ADH (arrows) and usual ductal hyperplasia (UDH) (H&E 20×).

**Figure 2 cancers-14-00507-f002:**
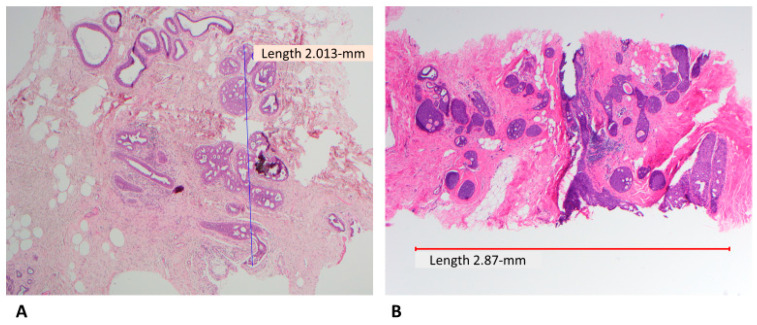
Severe ADH bordering on DCIS; (**A**) atypical intraepithelial ductal neoplasia arranged in cribriform pattern involving more than two spaces and measuring slightly more than 2 mm (H&E, scanning magnification); (**B**) Similar finding but with larger size, some pathologists may designate this as DCIS (H&E, scanning magnification).

**Figure 3 cancers-14-00507-f003:**
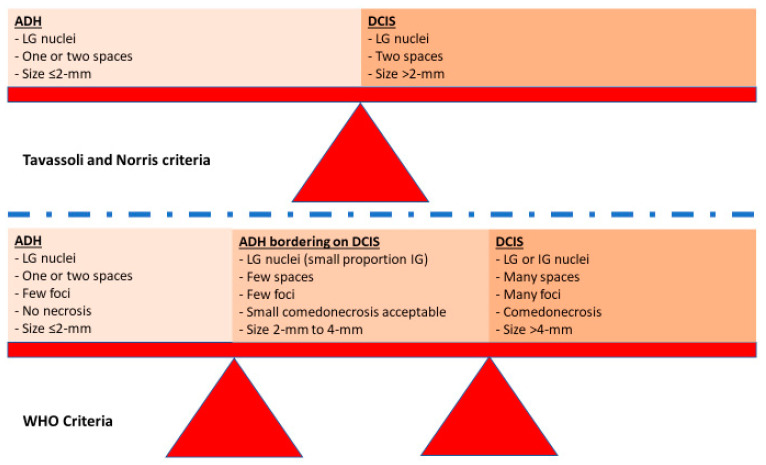
Schematic figure illustrating the difference between Tavassoli and Norris criteria and the WHO criteria.

**Figure 4 cancers-14-00507-f004:**
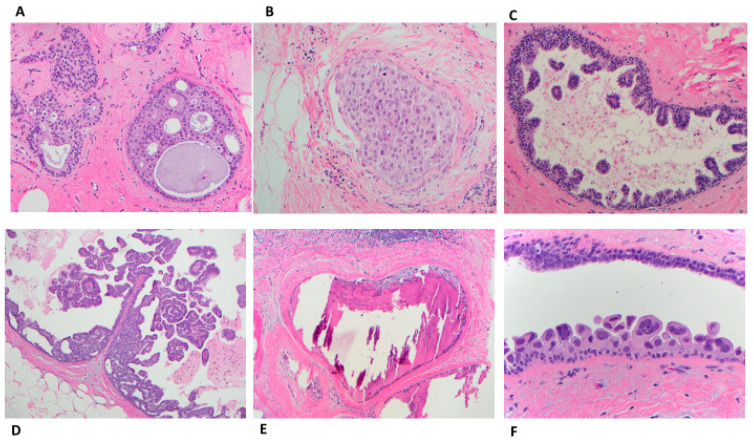
Ductal carcinoma in situ patterns; (**A**) cribriform with “cookie-cut” fenestrations (H&E 10×); (**B**) Solid (H&E 20×); (**C**) Micropapillary with club-shaped growth pattern (H&E 10×); (**D**) Papillary with fibrovascular core (H&E 10×); (**E**) Comedo with extensive intraductal expansile dirty necrosis and surrounding fibrosis and lymphocytes (H&E 10×); (**F**) clinging (flat) with disorderly arrange high grade cells (H&E 40×).

**Figure 5 cancers-14-00507-f005:**
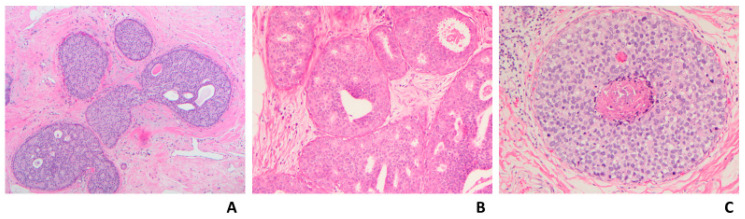
DCIS nuclear grades: (**A**) LG-DCIS: small monophonic cells arranged in a cribriform pattern (H&E, 10×); (**B**) IG-DCIS: ductal cells with moderate variability in size arranged in a cribriform pattern (H&E 20×); (**C**) HG-DCIS: Marked nuclear pleomorphism with many mitotic figures, apoptotic bodies and central necrosis arranged in a solid growth pattern (H&E 20×).

**Figure 6 cancers-14-00507-f006:**
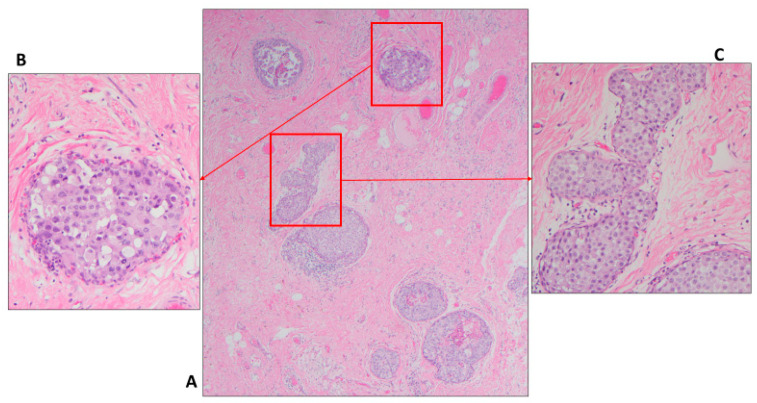
Heterogeneity of nuclear grade in the same section different ducts; (**A**) DCIS involving few ducts (H&E 4×); (**B**) duct with HG-DCIS (H&E 40×); (**C**) duct with IG-DCIS solid and cribriform pattern (H&E 40×).

**Figure 7 cancers-14-00507-f007:**
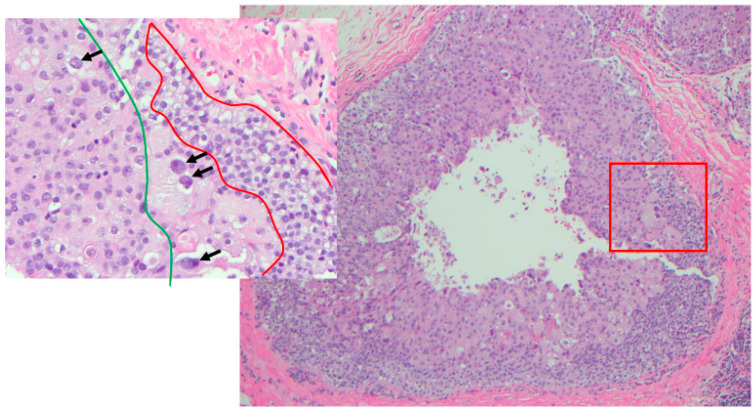
Heterogeneity of nuclear grade within the same duct (H&E 10×); in box LG (within red zone), IG (with green zone) and HG (arrows) (H&E 40×).

**Table 1 cancers-14-00507-t001:** Randomized active surveillance trials for low-risk DCIS.

	COMET	LORIS	LORD	LORETTA
Age	≥40	≥46	≥45	≥40; ≤75
Inclusion criteria	Imaging criteria: - Unilateral DCIS or bilateral if imaging ≤120 days of registration - When mammographic extent of calcifications exceeds 4 cm, a second biopsy benign or both sites fulfilling pathology eligibility criteria	Imaging criteria: - Screen-detected or incidental microcalcification	Imaging criteria: - Population-based screening mammography	Imaging criteria: - Lesion detected by mammography, US or MRI - Size ≤ 2.5-m
	Pathology Criteria: - ADH suspicious for DCIS - LG- or IG-DCIS - ER+ and/or PR+ - HER2− if performed Amended to include any degree of necrosis, or DCIS involving margin of excisional biopsy	Pathology Criteria: - LG-, IG-DCIS	Pathology Criteria: - Pure and LG- DCIS - Amended to include LG and IG DCIS; or patient preference Additional criteria: ER+ and HER2−	Pathology Criteria: - LG-, IG-DCIS ER+/HER2−
Exclusion criteria	- Males - HG-DCIS - Concurrent diagnosis of iBC - Bloody nipple discharge or skin changes - BI-RADS 4 or greater within 6 months - History of HT in the last 6 months	- Previous/current diagnosis of iBC or ipsilateral DCIS - Comedonecrosis - Mass lesion - Ipsilateral blood-stained nipple discharge - High risk group for developing breast cancer	- Personal history of DCIS or iBC - BRCA1 or BRCA2 in family - Symptomatic DCIS - Bilateral DCIS - Synchronous contralateral iBC, LCIS, Paget’s disease	- Comedonecrosis
Hormonal therapy	Optional	Not possible	Not possible	Mandatory
Design/Randomization	Two arms: Guideline concordant care +/− HT AS +/− HT	Two arms: Surgery Monitoring	Two arms: Standard treatment, surgery, +/− RT, +/− HT AS	Single arm
Follow-up	2, 5, and 7 years	10 years	10 years	5, 10 years
Endpoint	Primary: At 2, 5, 7 years: - Ipsilateral iBC rate Secondary: At 2 years: - Mastectomy/breast conservation rate - Contralateral iBC rate - Overall survival and disease-specific survival At 5, 7 years: - Overall survival and disease-specific survival	Primary: - Ipsilateral iBC free survival Primary outcome measure: - Ipsilateral iBC free survival time Secondary outcome measures: - Time to development of ipsilateral iBC, any iBC, contralateral iBC - Overall survival - Time to mastectomy, surgery	Primary: - Ipsilateral iBC-free percentage at 10 years Secondary: - Integrating clinical, imaging, morphological and molecular data to distinguish harmless from aggressive screen-detected DCIS	Primary: - Ipsilateral iBC-free percentage at 5 and 10 years

**Table 2 cancers-14-00507-t002:** List of studies simulating active surveillance trials for ADH or low-risk DCIS.

Author (Year) (Ref.)	Eligibility Criteria	Type of Upgraded to No. Upgraded/Total (%)	Risk Factors
Khoury (2019) [48]	COMET	iBC 2/124 (1.6) *	Not studied
		iBC 3/41 (7.3)	
Grimm (2017) [50]	COMET	DCIS to iBC 5/81 (6)	- Intermediate grade - ER- - PR-
	COMET	HG-DCIS 6/81 (7)	
	LORIS	iBC 5/74 (7)	
	LORIS	HG-DCIS 5/74 (7)	
	LORD	iBC 1/10 (10)	
	LORD	HG-DCIS 1/10 (10)	
Soumian (2013) [51]	LORIS	HG-DCIS 1/19 (5)	
		iBC 0/19 (0)	
Pilewskie (2016) [52]	LORIS	iBC 58/296 (20)	- Intermediate Grade - Mixed pattern architecture
Jakub (2017) [53]	LORIS	iBC 16/241 (6.6)	Not studied
Patel (2018) [54]	COMET	iBC 5/23 (22)	Not studied
	LORIS	iBC 6/25 (24)	Not studied
Oseni (2020) [55]	COMET	iBC 60/498 (12)	Not studied
	LORD	iBC 5/101 (5)	Not studied
	LORIS	iBC 38/343 (11.1)	Not studied
Zhan (2021) [23]	COMET	HG-DCIS and/or iBC 26/129 (20.2)	Span of mammographic calcifications (1.1 cm cutoff)
		iBC 12/129 (9.3)	- % space involved with comedonecrosis (53.5% cutoff) - Number of spaces with comedonecrosis
Khoury (2021) [56]	COMET	HG-DCIS 14/129 (10.9)	- Touching tumor infiltrating lymphocytes - Intermediate grade
Iwamoto (2021) [57]	LORIS	iBC 10/53 (19)	Not studied
	COMET	iBC 14/90 (16)	Not studied
	LORD	iBC 6/24 (25)	Not studied
	LORETTA	iBC 4/34 (12)	Not studied

* upgrade from ADH on core needle biopsy, the rest from DCIS.

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
