# Peer review of "Preneoplastic Low-Risk Mammary Ductal Lesions (Atypical Ductal Hyperplasia and Ductal Carcinoma In Situ Spectrum): Current Status and Future Directions"

_cancers, 2022, doi:10.3390/cancers14030507_

Round 1

Reviewer 1 Report

This review manuscript by Dr. Khoury provides an excellent and detailed overview of the about the current status of ADH and where we can go forward from here. The author presents a detailed analysis of the 4 clinical trials related to ADH/DCIS with pros and cons that can help in the development of new indications and assessment of ADH/DCIS. Also, the author compares ADH and DCIS defining how they relate and differentiate. Moreover, the author describes active surveillance trials, ductal carcinoma in situ, current treatments and gives a potential criterion to identify ADHs. Below I describe a few changes to strengthen this well-written manuscript:

Major points:

  1. Figures with H&E images: Please add more labels to panels so that they can stand alone and readers can easily identify what the author is describing. Two images with some arrows is not enough. Also, please label panels outside of the images. There is no reason to cover images with these letters.
  2. 2 has two panels A and B and Fig 3 hast 6 panels but they are not individually described in the text. Please describe and reference each panel in the text.
  3. Same comment as above applies for Figs. 5 and 6. Figures need better labels and each panel must be individually referenced in the text.
  4. Figure 4 legend needs to be a lot more descriptive and add details about the H7E images included.
  5. Please add scale bar two all of the figures containing H&E images.
  6. Rearrange figure 5: Images should not superimpose.
  7. Suggest adding a Venn diagram like figure to include how ADH and LG-DCIS compared to each other. Include the similarities and differences (maybe even add IG-DCIS and HG-DCIS). The text is very heavy in details (which is great!) so it would improve the manuscript to have figures with summaries of what is discussed in the text.
  8. Suggest adding a figure with a graphical description of what the author proposes as the criteria to identify ADH vs LG-DCIS.

Minor points:

  1. Abstract line 13: what is “these lesions” referring to?
  2. Paragraph in line 50: Introduce the different points in the order presented in the manuscript
  3. Fix text arrangement in tables so that everything is align (e.g. “Imaging criteria” in table 1)
  4. Figure 2: Labels of lesion lengths are unreadable. Please change.
  5. Suggest adding a list of acronyms at the beginning or end of the manuscript for readers to quickly reference as they read the text.
  6. Line 2343: Reference for Chavez de Paz Villanueva et al. Is missing in text. Please check all references.
  7. 2 legend: whats the magnification of images?

Author Response

Major points:

Figures with H&E images: Please add more labels to panels so that they can stand alone and readers can easily identify what the author is describing. Two images with some arrows is not enough. Also, please label panels outside of the images. There is no reason to cover images with these letters.

Response:

Labels are moved from the images and an illustrating image is added.

2 has two panels A and B and Fig 3 hast 6 panels but they are not individually described in the text. Please describe and reference each panel in the text.

Response:

Figure 2: both panels serve the same purpose which the size being larger than 2-mm, figure 2A the size is slightly larger than 2-mm and the lesion in figure 2B is larger. A and B are added in the text

Figure 3: Figure 3A to 3F are added in the text next to the corresponding morphology for clarity. Please these became figure 4

Figure 6: although they are two panels, the smaller panel is a higher magnification of the larger panel.  Please note this became figure 7

Same comment as above applies for Figs. 5 and 6. Figures need better labels and each panel must be individually referenced in the text.

Response:

Figure 5: Figure 5A, 5B and 5C are explained in the text. Please note these became figure 6

Figure 4 legend needs to be a lot more descriptive and add details about the H7E images included.

Response:

Figure 4: DCIS nuclear grades: A, LG-DCIS: small monophonic cells arranged in a cribriform pattern (H&E, 10X); B, IG-DCIS: ductal cells with moderate variability in size arranged in a cribriform pattern (H&E 20X); C, HG-DCIS: Marked nuclear pleomorphism with many mitotic figures, apoptotic bodies and central necrosis arranged in a solid growth pattern (H&E 20 X). please note this figure became figure 5

Please add scale bar two all of the figures containing H&E images.

Response:

I am afraid this is not possible as the pictures are already taken during my practice in the last 15 years. It is not possible to go back to the original slides to take new pictures with scale in them

Rearrange figure 5: Images should not superimpose.

Response:

Corrected

Suggest adding a Venn diagram like figure to include how ADH and LG-DCIS compared to each other. Include the similarities and differences (maybe even add IG-DCIS and HG-DCIS). The text is very heavy in details (which is great!) so it would improve the manuscript to have figures with summaries of what is discussed in the text.

Suggest adding a figure with a graphical description of what the author proposes as the criteria to identify ADH vs LG-DCIS.

Response:

Figure is added numbered 3

Minor points:

Abstract line 13: what is “these lesions” referring to?

Response:

“intraepithelial mammary ductal neoplastic lesions” replacing “these lesions”

Paragraph in line 50: Introduce the different points in the order presented in the manuscript

Response:

Fixed

Fix text arrangement in tables so that everything is align (e.g. “Imaging criteria” in table 1)

Response:

The inclusion criteria space is split into two spaces, one for the imaging and one for pathology

Figure 2: Labels of lesion lengths are unreadable. Please change.

Response:

Fixed

Suggest adding a list of acronyms at the beginning or end of the manuscript for readers to quickly reference as they read the text.

Response:

Added

Line 2343: Reference for Chavez de Paz Villanueva et al. Is missing in text. Please check all references.

Response:

It is mentioned by the end of the section (ref#59)

2 legend: whats the magnification of images?

Response:

Scanning magnification (using slide scanning image)

Reviewer 2 Report

Khoury's review aim was to evaluate ADH/DCIS as a spectrum of intraductal neoplastic disease (risk and histomorphology), examine the controversies of distinguishing DH versus DCIS and the grading of DCIS, review the upgrading for both ADH and DCIS with emphasis on the variation of methods of detection and the definitions of upgrading, and evaluate the impact of all these variables on the AS trials. I congratulate the author for an excellent review. The paper is very relevant for all who are dealing with diagnostics and treatment of non-palpable breast lesions. The paper should be published in the Cancers journal.

Minor remark: please define »UDH« in te Figure legends for Figure 1.

Author Response

Khoury's review aim was to evaluate ADH/DCIS as a spectrum of intraductal neoplastic disease (risk and histomorphology), examine the controversies of distinguishing DH versus DCIS and the grading of DCIS, review the upgrading for both ADH and DCIS with emphasis on the variation of methods of detection and the definitions of upgrading, and evaluate the impact of all these variables on the AS trials. I congratulate the author for an excellent review. The paper is very relevant for all who are dealing with diagnostics and treatment of non-palpable breast lesions. The paper should be published in the Cancers journal.

Response:

Thank you!

Minor remark: please define »UDH« in te Figure legends for Figure 1.

Response:

"Usual ductal hyperplasia” is added

Reviewer 3 Report

This review article covers the spectrum of low-risk ductal lesions and active surveillance trials. It is timely topic and of interest to readers. They systematically reviewed  and wrote well. I just would like to add  couple of minor comments.

  1. Page 2, line 93. ADH-overview --> It needs appropriate heading. a. ADH-overviw
  2. Page 3. line 131. First and Second for WHAT?
  3. Page 4, line 148. I guess you tried to say both "ADH underestimation" and "DCIS underestimation". For reader's understanding, please briefly describe the concept of both upgrading. 
  4. Generally, headings seem to be quite confusing. Introduction (concept and need for AS) --> pathologic issues of ADH and DCIS --> AS trials and simulating studies. I think this order would be easier to follow.

Author Response

This review article covers the spectrum of low-risk ductal lesions and active surveillance trials. It is timely topic and of interest to readers. They systematically reviewed  and wrote well. I just would like to add  couple of minor comments.

Page 2, line 93. ADH-overview --> It needs appropriate heading. a. ADH-overviw

Response:

“a” was omitted by the journal’s technical team

Page 3. line 131. First and Second for WHAT?

Response:

Meant challenges. The text is fixed

Page 4, line 148. I guess you tried to say both "ADH underestimation" and "DCIS underestimation". For reader's understanding, please briefly describe the concept of both upgrading.

Response:

This statement is added: Upgraded lesions are defined as lesions initially diagnosed as ADH in CNB but are found to harbor DCIS or iBC at excisional biopsy, or DCIS in CNB and are found to harbor iBC at the excisional biopsy.

Generally, headings seem to be quite confusing. Introduction (concept and need for AS) --> pathologic issues of ADH and DCIS --> AS trials and simulating studies. I think this order would be easier to follow.

Response:

I must admit that I had changed the layout few times, as each one of them has positives and negatives. One of these layouts is the one described by the reviewer. However, my main point here is how pathology plays a major role in the AS trials. Moreover, the upgrading which is part of ADH and DCIS has many information related to the AS trials. If I start with them, I will have difficulty referring to the AS since it would be mentioned later

For the technical team at Cancers, the layout of the manuscript is as follows:

  1. Introduction
  2. Active Surveillance Trials
  3. Atypical ductal hyperplasia and ductal carcinoma in situ spectrum-overview
    1. Atypical Ductal Hyperplasia-Overview
    2. Ductal Carcinoma in situ-Overview
    3. Severe Atypical Ductal Hyperplasia Bordering on Ductal Carcinoma in situ
  4. Upgrading
    1. Upgrading of Atypical Ductal Hyperplasia and Ductal Carcinoma in situ - Overall Studies
      1. Upgrading from ADH to Any Grade-DCIS and/or Invasive Carcinoma
      2. Upgrading from Ductal Carcinoma in situ to Invasive Carcinoma
    2. Upgrading of Atypical Ductal Hyperplasia and Ductal Carcinoma in situ - Studies Simulating Active Surveillance Trials
      1. Atypical Ductal Hyperplasia
      2. Ductal Carcinoma in situ
    3. Nomograms and Logistic Regression Analyses
      1. Atypical Ductal Hyperplasia
      2. Ductal carcinoma in situ
    4. Ductal Carcinoma in situ
      1. Histomorphology and Grading
      2. Natural History and Prognostic Tools for Ductal Carcinoma in situ
    5. Treatment
      1. Atypical Ductal Hyperplasia
      2. Ductal Carcinoma in situ
    6. Challenges in the Active Surveillance Trials from Pathology Standpoint
      1. Challenges in Grading Ductal Carcinoma in situ and their Consequences on Active Surveillance Trial
        1. Downgrading low-risk Ductal Carcinoma in situ to Atypical Ductal Hyperplasia or Atypical Ductal Hyperplasia bordering on Ductal Carcinoma in situ
        2. Low-Risk Ductal Carcinoma in situ Versus High-Grade Ductal Carcinoma in situ
      2. Comedonecrosis controversy and its consequences on active surveillance clinical trials:
    7. Conclusions